# Effects of Joint Lavage with Dimethylsulfoxide on LPS-Induced Synovitis in Horses—Clinical and Laboratorial Aspects

**DOI:** 10.3390/vetsci7020057

**Published:** 2020-04-30

**Authors:** Eric Danilo Pauls Sotelo, Cynthia Prado Vendruscolo, Joice Fülber, Sarah Raphaela Torquato Seidel, Fernando Mosquera Jaramillo, Fernanda Rodrigues Agreste, Luís Cláudio Lopes Correia da Silva, Raquel Yvonne Arantes Baccarin

**Affiliations:** 1Departamento de Clínica Médica, Faculdade de Medicina Veterinária e Zootecnia, Universidade de São Paulo, São Paulo 05508-270, Brazil; cynthiaimpoluto@hotmail.com (C.P.V.); jfulber@usp.br (J.F.); sarahseidel@usp.br (S.R.T.S.); ferjarmos@usp.br (F.M.J.); fe_nandara@hotmail.com (F.R.A.); 2Departamento de Cirurgia, Faculdade de Medicina Veterinária e Zootecnia, Universidade de São Paulo, São Paulo 05508-270, Brazil; silvalc@usp.br

**Keywords:** DMSO, equine, synovitis, joint inflammation

## Abstract

Several studies in human and equine medicine have produced controversial results regarding the role of dimethylsulfoxide (DMSO) as a therapeutic agent. This study aimed to evaluate the effect of joint lavage with different DMSO concentrations on biomarkers of synovial fluid inflammation and cartilage degradation in joints with lipopolysaccharide (LPS)-induced synovitis. Twenty-six tibiotarsal joints of 13 horses were randomly distributed into four groups (lactated Ringer’s solution; 5% DMSO in lactated Ringer’s; 10% DMSO in lactated Ringer’s; and sham). All animals were evaluated for the presence of lameness, and synovial fluid analyses were performed at 0 h, 1 h, 8 h, 24 h, and 48 h (T0, T1, T8, T24, and T48, respectively). The white blood cell counts (WBC), total protein (TP), urea, prostaglandin E2 (PGE2), interleukin (IL)-1β, IL-6, IL-10, tumor necrosis factor-α (TNF-α), hyaluronic acid (HA), and chondroitin sulfate (CS) concentrations were measured. The WBC counts and PGE2, IL-1β, IL-6, and TP concentrations increased in all groups at T8 compared to baseline values (*p* < 0.05). At T48, only the 5% DMSO and 10% DMSO groups showed a significant decrease in WBC counts (*p* < 0.05). Furthermore, the 10% DMSO group had lower concentrations of PGE2 and IL-1β at T48 than at T8 (*p* < 0.05) and presented lower IL-6 levels than the5% DMSO and lactated Ringer’s groups at T24. All groups showed an increase in CS concentration after LPS-induced synovitis. Joint lavage with 10% DMSO in lactated Ringer’s has anti-inflammatory but not chondroprotective effects.

## 1. Introduction

Dimethylsulfoxide (DMSO) is a liquid solvent with cryoprotective, analgesic, antimicrobial, and anti-inflammatory properties [1]. Although it has been known since the 19th century, when it was a by-product of the timber industry, its capacity to protect cellular systems from radioactivity as well as its cryopreservation effects were only reported for the first time in 1961 [2]. Since then it has been used in several laboratory and clinical studies, such as for the transport of compounds/molecules through tissue membranes, cellular cryopreservation, and local and systemic anti-inflammatory therapy, including septic disorders [3,4,5,6,7,8].

Degenerative joint disease or osteoarthritis is a common cause of economic loss and dysfunction in horses. Traumatic and inflammatory insults that upset articular cartilage homeostasis can initiate synovitis, and several studies suggest synovitis as a primary factor for osteoarthritis [9].

Injury to the synovial membrane alters its permeability and compromises its secretory functions, thus altering the synovial fluid properties. The synovial membrane is a source of direct release of prostaglandin E2 (PGE2), reactive oxygen species (ROS), metalloproteinases (MMPs), aggrecanases, and cytokines such as tumor necrosis factor-α (TNF-α) and interleukins (IL), which areincreased in response to injury, contributing to the articular cartilage degradation process.

DMSO is able to neutralize the effects of free radicals and suppress the production of PGE2, thereby inhibiting its cytotoxic effects [10,11,12]. Despite this, there are only a few studies evaluating the effects of DMSO on intra-articular structures, and studies regarding the use of DMSO as an adjunct to the treatment of synovitis and septic arthritis have reported conflicting results.

The safety of intra-articular DMSO injection on articular structures was first studied in the carpal joints of horses [13]. Clinical evaluation revealed no evidence of carpal inflammation after 40% DMSO solution was injected in the joints; on post-mortem examination, there was no difference in cartilage degradation between DMSO-treated joints and control jointson gross, histologic, or histochemical examinations. The study concluded that the use of intra-articular injections of DMSO did not produce adverse effects.

The intra-articular DMSO effects were also evaluated in a chemically-induced synovitis model in the carpal joints of horses [14]. Objective assessment of lameness did not reveal significant differences between treatment (40% DMSO in lactated Ringer’s solution) or control limbs (lactated Ringer’s solution), and no difference was detected in histochemical staining of articular cartilage. However, the DMSO-treated group did have lower synovial WBC counts and subjective synovial membrane inflammatory scores than the controls.

Joint lavage with DMSO was investigated in a study using healthy carpus and tarsus joints of horses. The joints were lavaged once with either lactated Ringer’s, 10% DMSO in lactated Ringer’s, or 30% DMSO in lactated Ringer’s. The results of this study showed that lactated Ringer’s, 10% DMSO, and 30% DMSO induced similar inflammatory changes, which were significantly greater than those induced by arthrocentesis alone [15].

Nevertheless, Smith et al. [16] showed a decrease in the synovial WBC count, especially neutrophils, when a topical gel containing DMSO (15g; 90%) was applied every 12 h for 60 h in an lipopolysaccharide (LPS)-induced synovitis model in the middle carpal joints of horses. No other studies regarding the use of DMSO as a topical anti-inflammatory exist in the literature.

In vitro studies showed that DMSO protected the synovial fluid from degradation [17]. However, Smith et al. [18] observed that concentrations of DMSO equal to or in excess of 5% suppressed articular cartilage matrix metabolism. Additionally, the authors cautioned that adding DMSO to intra-articular lavage solutions may have deleterious effects on cartilage metabolism. Moses et al. [19] showed that 10% DMSO solution failed to suppress prostaglandin E2 production.

Although DMSO is used frequently in equine veterinary medicine [20,21], particularly for joint lavages for the treatment of synovitis or osteoarthritis, there is weak evidence to support its beneficial use as an anti-inflammatory agent. This is probably because few studies focused on the effects of DMSO on joint biomarkers. Moreover, there is no consensus on the best concentration of DMSO to obtain beneficial intra-articular anti-inflammatory and antioxidant effects without extracellular matrix catabolism.

The objective of this study was to evaluate the anti-inflammatory and chondroprotective effects of different DMSO concentrations in lactated Ringer’s solutions on LPS-induced synovitis in horses.

## 2. Materials and Methods

### 2.1. Experimental Design

The experimental protocol used in this study was approved by the Animal Ethics Committee of the School of Veterinary Medicine and Animal Science, University of São Paulo (protocol number: 9265010917; date of approval: 16/07/2018) and was carried out in accordance with the U.K. Animals Scientific Procedures Act, 1986 and associated guidelines, and the EU Directive 2010/63/EU for animal experiments.

The study included 26 tibiotarsal joints of 13 healthy male and female horses, with an average age of five years and weights ranging from 350 to 400 kg, with no history of joint diseases. During the experimental period, the horses were kept in single 12 m² stalls (3 × 4 m) and fed pellets (1% of the animal’s body weight); coast cross hay and water were offered ad libitum.

All horses were evaluated for lameness, and the normality of the joints was determined by radiography and ultrasonography according to Silva et al. [22]. For the radiographic evaluation, the lateromedial, dorsoplantar, dorsolateral–plantaromedial oblique, and dorsomedial–plantarolateral oblique views of both tarsi were performed (Poskom PXP 20HF X-Ray; Fujifilm, model CC). Ultrasonographic examinations (ESAOTE MyLab 30 VET Ultrasound system, linear multi-frequency transducer 7.5 to 12 MHz, ESAOT, Italy) were carried out in B and Power Doppler mode. All joint surfaces were submitted to longitudinal and cross-sectional imaging with the joint in the weight-bearing and flexed position.

The 26 tibiotarsal joints were assigned randomly to four groups in a way that the same horse did not receive the same treatment in both joints. The groups were treated and organized as follows: group A—lactated Ringer’s solution (n = 7); group B—5% DMSO in lactated Ringer’s solution (n = 7); group C—10% DMSO in lactated Ringer’s solution (n = 7); and group D—sham (n = 5).

### 2.2. Sample Size Calculation

The sample size was calculated to achieve 80% statistical power for detecting a 30% difference in interleukin IL-1 and chondroitin sulfate (CS) levels between the groups with a 2-sided alpha level of 0.025 and β = 0.20 based on 2-way analysis of variance (ANOVA).

### 2.3. Randomization

Four conditions (treatments) and two joints for each horse (2 joints) were tested; therefore, a randomized block design was used. To avoid order effects for each treatment, as well as to have an equal number of repetitions, 12 cards were made with the possible pairs of groups and the order in which the tibiotarsal joints (Right/Left or Left/Right) would be treated. The cards were kept in a box, and for each horse, the card was drawn at the time of the treatment to address concealment. The investigator was thus blinded to the treatment allocation and the joint that would be treated first until the time of the first injection (Table 1).

### 2.4. Induction of Synovitis and Joint Lavage

Initially, intra-articular injections of sterile phosphate buffered saline (PBS, 2 mL) containing 0.5 ng of *Escherichia coli* LPS (from *E. coli* O55:B5, catalog #L5418; Sigma-Aldrich, USA) were administered to one joint of each animal at time 0 (T0). Only one tibiotarsal joint of each animal was treated at a given time. The interval between the treatments of the two tibiotarsal joints of the same animal was at least three weeks.

Before LPS administration, synovial fluid (SF) samples were collected (T0) for the baseline determinations. Arthrocentesis was performed medially to the saphenous vein, just below the medial malleolus of the tibia, in the dorsomedial face of the tarsus. The site was trichotomized and sterilized with 2% chlorhexidine detergent and 0.5% alcoholic chlorhexidine.

After 8 h, the designated treatment was performed, i.e., joint lavage with lactated Ringer’s solution, 5% DMSO in lactated Ringer’s solution, or 10% DMSO in lactated Ringer’s solution.

Two accessions, dorsomedial and dorsolateral, were used for joint lavage with 18G hypodermic needles. The first access was also used for SF collection.

The animals were assessed three times a day for any signs of discomfort.

### 2.5. Lameness Evaluation

All horses were evaluated prior to the collections and at 0 h, 1 h, 8 h, 24 h, and 48 h (T0, T1, T8, T24 and T48, respectively) for the presence of joint pain-induced lameness. The presence of lameness in horses was objectively evaluated, using Lameness Locator^®^, (Equinosis, Columbia, MO, USA) which is capable of indicating the source of the lameness and quantifying it. The equipment is able to dynamically quantify the lameness by noninvasive wireless sensors, one placed on the head between the ears on the frontal bone (accelerometer) and the other in the dorsal portion of the pelvis between the ischiatic turberosities (accelerometer), and a gyroscopic sensor placed on the dorsal face of the right thoracic limb. Then, in real time, a program converts the sensor information into vector form, and the intensity of the claudication is given by a Q-score resulting from the calculation of variations in movements. For pelvic limbs, the total range of pelvic movement may not exceed ±3.

### 2.6. Synovial Fluid Analysis

SF samples were collected aseptically at T0 (before LPS injection), T1, T8, T24, and T48 after LPS injection. All collections were performed by dorsomedial access medially to the saphenous vein.

An aliquot (500 µL) of the SF was processed immediately for the white blood cell (WBC) count. The remaining SF (3–4 mL) was centrifuged at 2000× *g* for 15 min at 4 °C, and the supernatant was aliquoted and stored at −80 °C for analyses of total protein (TP), PGE2, cytokines, and glycosaminoglycans. To compensate for the possible dilution of the SF samples, urea concentrations were measured in terms of urease–glutamate dehydrogenase using an automated biochemical analyzer (Randox, Crumlin, UK) [23].

The WBC counts were performed in a Neubauer chamber using in natura SF aliquots. Cytologic examination of SF was carried out on smears made from centrifuged specimens. The smears were stained with Rosenfeld dye. The TP and urea levels in the SF were measured using the biuret method with an automated biochemical analyzer.

PGE2 levels in the SF weremeasured using a commercial enzyme-linked immunosorbent assay (ELISA) kit (Cayman Chemical, Ann Arbor, MI, USA).

IL-1β, IL-6, IL-10, and TNF-α levels were quantified using an equine cytokine/chemokine panel based on Luminex technology. SF samples (200 µL) were previously treated with bovine testis type 1 hyaluronidase enzyme (10 µL; 100 U/mL in 0.05 M acetate buffer) (Sigma-Aldrich Brasil Ltda., Sao Paulo, Brazil). All determinations were performed in duplicate.

Hyaluronic acid (HA) and CS concentrations were determined as previously described by Machado et al. [24]. In brief, SF samples (100 µL) were submitted to proteolysis (4 g/L maxatase in 0.05 M Tris-HCl, pH 8.0, 200 µL). After incubation overnight at 50 °C, maxatase was heat inactivated, and the debris was removed by centrifugation. The supernatant was collected, frozen, dried, and resuspended in 50 µL of water. The identification of SF glycosaminoglycans was performed by a combination of agarose gel electrophoresis (0.55%) in a 0.05 M 1,3-diaminopropane-acetate (PDA) buffer, pH 8.0 and differential staining of sulfated and non-sulfated glycosaminoglycans by Toluidine Blue at different pH [23]. These compounds were quantified by densitometry of the electrophoresis gel slabs.

### 2.7. Statistical Analysis

The data obtained were evaluated for normality by the Kolmogorov–Smirnov test. The variables were analyzed using the 2-way ANOVA model considering the treatment factors, time, and interaction between both. Post-hoc analysis was performed using the Tukey–Kramer test or Kruskal–Wallis non-parametric test for variables that did not show a normal distribution. The significance level was set to 5% (*p* < 0.05).

## 3. Results

### 3.1. Lameness Evaluation

Lameness was observed in 50% of the animals 8 h after the induction of synovitis (T8): 2/7 in the lactated Ringer’s group, 4/7 in the 5% DMSO group, 4/7 in the 10% DMSO group, and 3/5 in the sham group. After 24 h (T24), 27% of the animals presented lameness: 2/7 in the lactated Ringer’s group, 3/7 in the 10% DMSO group, and 2/5 in the sham group; no animal presented lameness in the 5% DMSO group. At T48, only 19% of the animals still had lameness: 2/7 in the lactated Ringer’s group, 1/7 in the 5% DMSO group, and 2/7 in the 10% DMSO group. There was no statistically significant difference in the mean sum of vectors among groups or moments (*p* > 0.05) (Figure 1A).

### 3.2. White Blood Cell Count of the Synovial Fluid

The SF exhibited increased WBC counts in all groups 8 h after LPS injecttion (T8) compared to those at baseline(T0) and one hour after injection (T1) (*p* < 0.05). WBC counts decreased slowly (24 and 48 h), and 5% DMSO and 10% DMSO groups showed a significant decrease at T48 in relation to T8 (*p* < 0.05). There was no statistically significant difference among the groups (*p* > 0.05) (Figure 1B). There was a predominance of polymorphonuclear cells in the lactated Ringer´s, 10% DMSO, and sham groups at T24 and T48 compared to that at T0, T1, and T8 (*p* < 0.05). At T48, the group treated with 10% DMSO showed a higher percentage of polymorphonuclear cells than the lactated Ringer´s, 5% DMSO, and sham groups (*p* < 0.05).

### 3.3. Total Protein and UreaConcentrations in the Synovial Fluid

All groups exhibited significantly increased concentrations of TP at T8, T24, and T48 (*p* < 0.01) compared to those at T0, but the TP concentrations were not significantly different among the groups (*p* > 0.05) (Figure 1C). Regarding urea concentrations, there was no significant difference among the groups (*p* > 0.05) (Figure 1D).

### 3.4. Prostaglandin E2 in the Synovial Fluid

Compared to the corresponding baseline values (T0), all groups exhibited an increase in the PGE2 concentration in the SF at T8 (*p* < 0.05). Lactated Ringer´s and sham groups also showed an increase in PGE2 concentrations at T1 than those at baseline (*p* < 0.05). These values decreased with time and reached baseline values at T48. Only the 10% DMSO group presented a significant decrease at T48 relative to T8 (*p* < 0.05); however, there was no significant difference among the groups (*p* > 0.05) (Figure 2).

### 3.5. InterleukinLevels in the Synovial Fluid

LPS caused an increase in the IL-1β and IL-6 concentrations in all groups at T8 compared to the corresponding baseline values (*p* < 0.05), and also in the IL-6 concentration relative to the T1 values. Only the 10% DMSO group exhibited a significant decrease in the IL-1β concentration at T48 compared to that at T8 (*p* < 0.05). Additionally, the 10% DMSO group showed a lower concentration of IL-6 at T24 than the 5% DMSO and the lactated Ringer’s group (Figure 3A,B).

Although IL-10 concentration in the lactated Ringer´s group increased at T1 and T48 relative to baseline values, only the 10% DMSO group exhibited an increase in the IL-10 concentration at T48 compared to that at T0, T1, T8, and T24 (*p* < 0.05) (Figure 3C).

All groups exhibited an increase in TNF-α concentration one hour after the LPS injection (T1) (*p* < 0.05). The group treated with 10% DMSO and the sham group presented a significant decrease in TNF-α concentration at T24 and T48 compared to that at T1 (*p* < 0.05) (Figure 3D).

### 3.6. Glycosaminoglycans in the Synovial Fluid

All groups demonstrated an increase in the CS concentration either at T24 or T48 or both (*p* < 0.001) compared to that at baseline, T1, and T8 (Figure 4A). The lactated Ringer’s and 10% DMSO groups presented a decrease in the HA concentration at T24 compared to that at T1, and the 10% DMSO group also showed a decrease in the HA concentration relative to the baseline value (*p* < 0.01) (Figure 4B).

## 4. Discussion

Acute and transient inflammation was induced in equine joints by an intra-articular injection of 0.5 ng of LPS, which is a well-described experimental model. Our results were similar to other studies [25,26,27]; that is, the effects observed were limited to the joint environment, with no systemic consequences for the horses.

Early evidence of synovitis was demonstrated by the Lameness Locator, which revealed an abnormal gait in all groups 8 h after the LPS injection. This system is more sensitive in detecting lameness than a subjective evaluation, even when carried out by experienced veterinarians, and produces quantitative results [28]. The lameness almost disappeared in all groups at 48 h, and there was no statistical difference among them.

In the present study, the WBC counts were significantly higher 8 h after the LPS injection in all groups compared to normal values by up to 1000 cells/µL [29] and decreased thereafter. Only in the 5% DMSO and 10% DMSO groups, the WBC counts significantly decreased at T48 compared to those at T8, however the values were still above normal. Similar to the WBC results, TP concentrations increased at T8 in all groups; however, unlike the WBC counts, the TP concentrations remained high throughout the study. These findings corroborate those of other studies that used DMSO for the treatment of experimentally-induced synovitis and for joint lavage in healthy equines [13,14,15].

PGE2, IL-10, and TNF-α concentrations increased after one hour of LPS-induced synovitis in the present study, however, PGE2 had a higher peak concentration after 8 h. Other studies showed that PGE2 concentrations increase between 2 h and 12 h [30,31].

The joint lavage with 10% DMSO in lactated Ringer´s solution decreased PGE2 concentration after 48 h of the LPS-induced synovitis. This was an expected outcome, which can be explained by the anti-inflammatory effects of DMSO demonstrated by several other authors [4,8,25]. In contrast, Moses et al. [19] showed that a 10% DMSO solution failed to suppress prostaglandin E2 production.

The studies that evaluated the use of DMSO in equine joints do not report its effect on inflammatory cytokines. DMSO causes a dose-dependent inhibition of superoxide radicals produced by stimulated neutrophils, and the clinical benefit of DMSO can be attributed to its free radical clearing capacity [32]. In vitro and in vivo studies indicate that DMSO is capable of reducing levels of pro-inflammatory cytokines and lymphocyte activation [8,33]. In the present study, the treatment with 10% DMSO in lactated Ringer´s solution decreased IL-1β concentrations at 48 h compared to those at 8 h (*p* < 0.05) and also lowered IL-6 levels compared to the 5% DMSO and lactated Ringer’s treatments at T24.

The analysis of TNF-α concentrations in the SF can anticipate radiographic lesions in joints with osteoarthritis [34]. As observed in this study, after one hour of synovitis induction, all groups showed a significant increase in TNF-α concentration. TNF-α is present in acute inflammatory processes, and its presence is related to signage of synoviocytes and chondrocytes, inciting the release of other molecules responsible for joint degradation [25]. It was not possible to observe the influence of DMSO on the behavior of this cytokine.

IL-10 is a product of T helper (Th2) cells, B cells, and monocytes. This cytokine has immunoregulatory and anti-inflammatory effects that inhibit the synthesis of IL-1, IL-6, and IL-8 and contribute toward reducing joint cartilage damage [35]. IL-10 levels increased significantly after LPS-induced synovitis; however, it was not possible to conclude that the treatments increased the expression of this cytokine.

Glycosaminoglycans are efficient biomarkers of joint metabolism, which may indicate the degradation of the cartilage matrix [24]. In the present study, evaluation of glycosaminoglycans in the SF indicated a breakdown of HA molecules and increased CS concentration; however, HA values had already returned to baseline values at the end of the study. As the intra-articular DMSO use did not inhibit the CS build up in synovial fluid, it was not possible to demonstrate chondroprotective effects.

There is no consensus about the effects of intra-articular DMSO use among researchers. An in vitro study of the effects of DMSO at different concentrations and periods of exposure indicated a dormancy effect on chondrocytes and consequent suppression of metabolism, but without death of chondrocytes [18]. Another study showed low toxicity of DMSO in human chondrocytes, andglycosaminoglycan production was not affected [7]. However, in our study, the chondroprotective effect of DMSO was not demonstrated.

## 5. Conclusions

Regarding joint lavage in horses, the use of 10% DMSO in lactated Ringer´s solution has more beneficial anti-inflammatory effects than the use of lactated Ringer’s only or 5% DMSO in lactatedRinger´s solution. However, none of the treatments demonstrated a chondroprotective effect.

## Figures and Tables

**Figure 1 vetsci-07-00057-f001:**
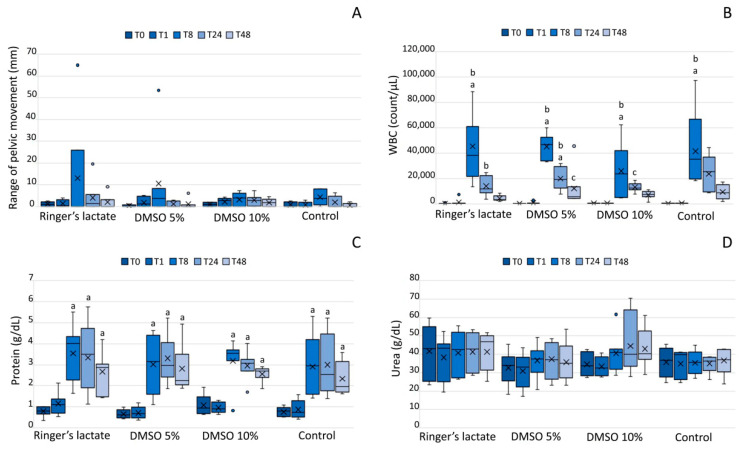
(**A**) Lameness evaluation (Lameness Locator), (**B**) white blood cell (WBC) counts (cell/μL), (**C**) total protein concentrations (g/dL), (**D**) urea concentrations (g/dL) in synovial fluid. Lameness evaluation (Lameness Locator), WBC counts, and total protein and urea concentrations are shown as box plots, indicating the median (−), mean (×), first and third quartiles. ^a^ Statistically significant differences compared to the baseline values (*p* < 0.05); ^b^ Statistically significant differences compared to 1 h post-induction of synovitis (T1) (*p* < 0.05); ^c^ Statistically significant differences compared to T8 (*p* < 0.05). DMSO: dimethylsulfoxide.

**Figure 2 vetsci-07-00057-f002:**
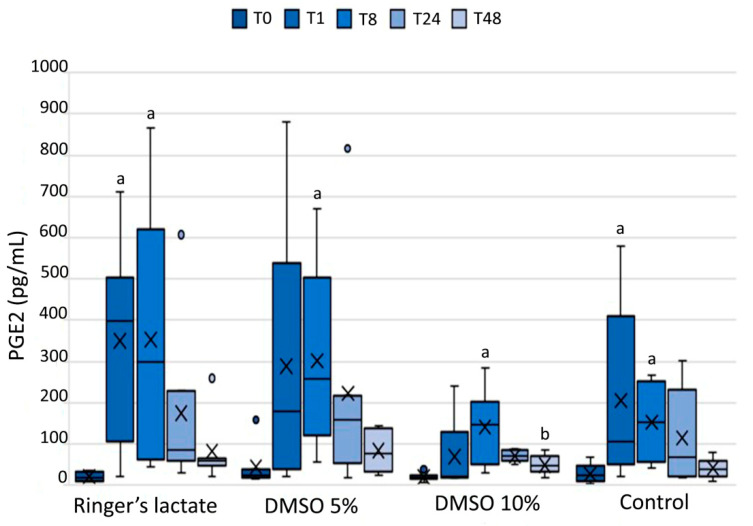
Prostaglandin E2 (PGE2) concentration (pg/mL) in the synovial fluid. PGE2 concentrations are shown as box plots, indicating the median (−), mean (×), and first and third quartiles. ^a^ Statistically significant differences compared to the baseline values (*p* < 0.05); ^b^ Statistically significant differences compared to T8 (*p* < 0.05).

**Figure 3 vetsci-07-00057-f003:**
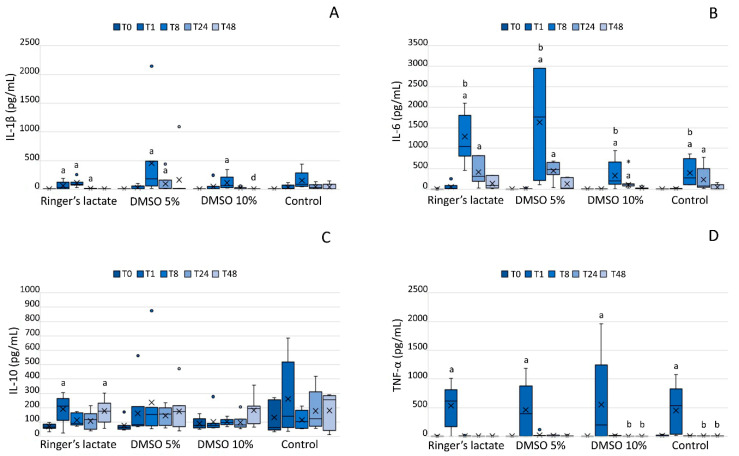
Interleukin (IL)-1β (**A**), IL-6 (**B**), IL-10 (**C**), and tumor necrosis factor (TNF)-α (**D**) concentrations (pg/mL) in the synovial fluid. IL-1β, IL-6, IL-10, and TNF-α concentrations (pg/mL) in the synovial fluid are shown as box plots, indicating the median (−), mean (×), and first and third quartiles. ^a^ Statistically significant differences compared to the baseline values (*p* < 0.05); ^b^ Statistically significant differences compared to T1 (*p* < 0.05); ^d^ Statistically significant differences compared to T8 (*p* < 0.05). ***** Statistically significant differences compared to the 5% DMSO and the lactated Ringer’s group (*p* < 0.05).

**Figure 4 vetsci-07-00057-f004:**
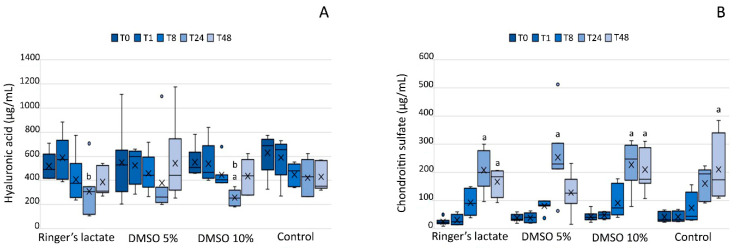
(**A**) Hyaluronic acid and (**B**) chondroitin sulfate concentrations (μg/mL) in the synovial fluid. Hyaluronic acid and chondroitin concentrations are shown as box plots, indicating the median (−), mean (×), and first and third quartiles. ^a^ Statistically significant differences compared to the baseline values, T1 and T8 (*p* < 0.05). ^b^ Statistically significant differences compared to T1 (*p* < 0.05).

**Table 1 vetsci-07-00057-t001:** Randomization of the sequence cards.

Card	Right Tibiotarsal Joint	Left Tibiotarsal Joint	Card	Right Tibiotarsal Joint	Left Tibiotarsal Joint
1	A	B	8	A	C
2	C	D	9	C	A
3	B	A	10	D	B
4	D	C	11	C	B
5	B	C	12	A	C
6	A	D	13	B	A
7	B	D			

A: lactate Ringer’s solution; B: 5% DMSO in lactate Ringer’s solution; C: 10% DMSO in lactate Ringer’s solution; D: sham.

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
