# Peer review of "Effects of Joint Lavage with Dimethylsulfoxide on LPS-Induced Synovitis in Horses—Clinical and Laboratorial Aspects"

_vetsci, 2020, doi:10.3390/vetsci7020057_

Round 1

Reviewer 1 Report

The manuscript reads smoothly.

a few points that possibly improve the manuscript:

line 83 "the effects" it is not clear at this point which biological endpoint are being evaluated

line 1380148, is there any correction needed for the fact that the skin is loosely connected to the boden ant therefor the sensor under/over estimate the real movement at the boen level?

Was there any kind of correction for multiple measurements?

It took me a while before I could easily read the figures 

line 310 "beneficial effects" related to inflammation only and not the chondroprotective, this is sated  in the flooring sentence. but consider to add beneficial inflammatory .

in the references list numerous page numberings are lacking e.g. ref 14, 15, 16, 20, 21, 23, 28 etc

ref 25 has capitals 

Author Response

We thank the reviewers’ comments that greatly improved our paper, and hope that the modifications introduced will now meet your requirements.

Point 1: The manuscript reads smoothly.

A few points that possibly improve the manuscript:

line 83 "the effects" it is not clear at this point which biological endpoint are being evaluated

Response 1: This information was included in the Introduction, page 2, line 83.

Point 2: line 1380148, is there any correction needed for the fact that the skin is loosely connected to the boden ant therefor the sensor under/over estimate the real movement at the boen level?

Response 2: No. The Lameness Locator device can be analysed without corrections. Some papers describe how measurements are made.

Keegan KG1, Yonezawa Y, Pai PF, Wilson DA, Kramer J. Evaluation of a sensor-based system of motion analysis for detection and quantification of forelimb and hind limb lameness in horses. Am J Vet Res. 2004 May;65(5):665-70.

Keegan KG1, Kramer J, Yonezawa Y, Maki H, Pai PF, Dent EV, Kellerman TE, Wilson DA, Reed SK. Assessment of repeatability of a wireless, inertial sensor-based lameness evaluation system for horses. Am J Vet Res. 2011 Sep;72(9):1156-63. doi: 10.2460/ajvr.72.9.1156.

McCracken MJ1, Kramer J, Keegan KG, Lopes M, Wilson DA, Reed SK, LaCarrubba A, Rasch M. Comparison of an inertial sensor system of lameness quantification with subjective lameness evaluation. Equine Vet J. 2012 Nov;44(6):652-6. doi: 10.1111/j.2042-3306.2012.00571.x.

Also, there is some information about how Lameness Locator is measured at https://equinosis.com/veterinarians/

Point 3: Was there any kind of correction for multiple measurements?

Response 3: Yes. To compensate for the possible dilution of the synovial fluid samples, urea concentrations were measured. However no significant fluctuations in the urea concentration were observed and the concentrations of protein, PGE2, cytokines, HA and CS were not presented as urea ratios.

Kraus VB, Stabler TV, Kong SY, Varju G, McDaniel G. Measurement of synovial fluid volume using urea. Osteoarthritis Cartilage 2007;15:1217-1220.

Kraus VB, Huebner JL, Fink C, King JB, Brown S, Vail TP, Guilak F. Urea as a passive transport marker for arthritis biomarker studies. Arthritis Rheum 2002;46:420-427.

Point 4: It took me a while before I could easily read the figures

Response 4 - All information was inserted in the figure according to instructions. We agree that this information makes it difficult to read the figures quickly.

line 310 "beneficial effects" related to inflammation only and not the chondroprotective, this is sated  in the flooring sentence. but consider to add beneficial inflammatory .

Response 4: done

Point 5: in the references list numerous page numberings are lacking e.g. ref 14, 15, 16, 20, 21, 23, 28 etc

ref 25 has capitals  

Response 5: done

Reviewer 2 Report

The paper is interesting and well written, but does not add much on the behavior of cytokines following treatment with DMSO.
The antioxidant properties of DMSO are quite well known and the result obtained by the authors does not differ from this assumption.

I would ask the authors to clarify in discussion why 10% DMSO does not have chondroprotective actions and why it does not interfere, despite the antioosidant properties, with the cytokine profile.

Author Response

We thank the reviewers’ comments that greatly improved our paper, and hope that the modifications introduced will now meet your requirements.

Point 1: The paper is interesting and well written, but does not add much on the behavior of cytokines following treatment with DMSO. The antioxidant properties of DMSO are quite well known and the result obtained by the authors does not differ from this assumption.

Response 1:  

The studies that evaluated the use of DMSO in equine joints do not report its effect on inflammatory cytokines. Besides that, studies regarding the use of DMSO as an adjunct to the treatment of synovitis and septic arthritis have reported conflicting results.

In the present study, we showed that DMSO decreased IL-1β and IL-6 concentrations in synovial fluid, and we can safely indicate the use of 10% DMSO in lactated Ringer´s solution for joint lavage in horses because it has more beneficial anti-inflammatory effects

Point 2: I would ask the authors to clarify in discussion why 10% DMSO does not have chondroprotective actions …..

Response 2:

The text was improved (line 305-307).

Explanation:

Changes in PG and glycosaminoglycan (GAG) structure and concentration result in changes of compressive stiffness and contribute to cartilage damage. We have shown that synovial fluid CS indicates abnormal joint metabolism in osteochondritis dissecans (1), and that CS is a good biomarker to evaluate the cartilage turnover and homeostasis(2).

Machado TSL, Correia da Silva LCL, Baccarin RYA, Michelacci YM. Synovial fluid chondroitin sulphate indicates abnormal joint metabolism in asymptomatic osteochondritic horses. Equine Vet J. 2012;44: 404–411.

Baccarin RYA, Rasera L, Machado TSL, Michelacci YM. Relevance of synovial fluid chondroitin sulphate as a biomarker to monitor polo pony joints. Can J Vet Res. 2014;78: 50–60.

In the present study all groups demonstrated an increase in the CS concentration either at T24 or T48 or both (P<0.001) compared to that at baseline. DMSO didn´t inhibited the CS build up in synovial fluid. So, we couldn´t affirm that DMSO was more efficient to preserve the cartilage, i.e , it didn´t show chondroprotective effects in this study.

and why it does not interfere, despite the antioosidant properties, with the cytokine profile.

Response 2 : In the present study, the treatment with 10% DMSO in lactated Ringer´s solution decreased IL-1β concentrations at 48 hours compared to those at 8 hours (P<0.05) and also lowered IL-6 levels compared to the 5% DMSO and lactated Ringer’s treatments at T24. So, in the present study DMSO interfered with cytokine profile.